# Supporting Resilient Urban Planning through Walkability Assessment

**Francesca Abastante** , **Isabella M. Lami** , **Luigi La Riccia** **and Marika Gaballo** *

Interuniversity Department of Regional and Urban Studies and Planning (DIST), Politecnico di Torino,
Viale Mattioli 39, 20125 Turin, Italy; francesca.abastante@polito.it (F.A.); isabella.lami@polito.it (I.M.L.);
luigi.lariccia@polito.it (L.L.R.)
* Correspondence: s251715@studenti.polito.it

**Abstract:** The urban planning and evaluation literature suggests that making a walkable city means creating a resilient and healthy city. In recent years, alternative mobility has been the subject of numerous studies, showing that the concept of urban walkability can be used as an additional support in planning resilient cities. Though researchers agree that walkability assessment has a positive impact on public space planning, it is still difficult to include the topic in planning strategies because of its novelty in the scientific debate. This paper will first review the literature on walkability assessment and then propose a multi-methodological assessment framework that fills the gaps in existing assessment methods. The multi-methodological assessment framework contributes to overcoming the idea that objective and subjective aspects are "not part of the same planning project." Thanks to its combination of hard and soft methods, the assessment framework illustrated in this paper can consider physical and perceptual aspects simultaneously and represent them visually using Geographic Information Systems (GIS). It can thus provide easily readable results that can be applied in establishing guidelines for planning resilient cities.

**Keywords:** walkability; walkability measure; urban resilience; quantitative; qualitative and mixed models and methods; urban planning; public space

## 1. Introduction

*Let's think about how our cognitive ability and our experience will diminish, for example looking at the use of Google Maps: well, people have no idea where it is interesting to walk because they are glued to the phone to get in the most efficient way from A to B. More an experience is smooth, without clutches, more we stop learning* [1] (p. 36).

The concept of resilience originated in ecology and refers to a "measure of persistence of systems and of their ability to absorb change and disturbance and still maintain the same relationship between population or state variables" [1]. The concept has gained increasing importance in numerous disciplines [2] including urban planning, [3,4] where many researchers have increasingly stressed the need for tools to support appropriate policies for creating resilient and inclusive cities [5–7].

The "city object" can be considered as a rather complex "urban ecosystem" that is vulnerable to change and external inputs [8], and needs conceptual and operational models to support its development and stability [9]. Accordingly, the challenge of urban planning is to design adaptive settlements capable of facing the threats to resilience [10]. In this perspective, the term resilient is not used to design or describe an ideal urban space [11] but to emphasize the need for urban spaces able to be safe, livable, open, accessible, healthy and designed to a human scale [12–15].

Urban mobility is one of the most significant aspects of the complex challenge that cities are facing in the areas of sustainability and climate change resilience [11,16,17]. In particular, reducing

car-dependency can significantly reduce the negative impact of neighborhoods in terms of emissions and energy consumption: walkable neighborhoods can assist in climate change mitigation and adaptation plans by decreasing reliance on fossil fuels used for transportation [18].

According to this, the scientific communities and public administrations are now called on to identify development models that can reduce pollutant emissions by improving "soft mobility" [19]. As the easiest, cheapest, and socially most equal form of soft mobility, walkability presents a variety of advantages: economic, political (saving non-renewable resources), social (equity of mobility), and ecological. [20,21].

Researchers agree that walkability is first of all a measurement tool for assessing the degree of pedestrian use of a certain area [22]. It is important to stress that there is still no consensus definition of walkability [23]: some scholars define it as "the security, economy, and convenience of traveling by foot" [24], while others adopt a more qualitative perspective, regarding walkability as a "quality of place" [25].

Those differences in the definition of walkability are due to several factors. First, the action of "walking" is ambiguous: people walk for many reasons and it is difficult to determine whether walkability planning should be classified as a matter of security, health, or transport [26]. Second, walkability affects multiple stakeholders, aspects, and different spheres of reality. Third, walkability can be analyzed and measured at different territorial scales [27]. Lastly, and maybe more important, the broader concept of walkability includes a wide range of subjective elements (comfort, continuity, legibility) that are often difficult to interpret [28–30]. Moreover, subjectivity could be understood as the relationship between the perception of a space's quality and the reaction that this space is able to generate in the observer. This relationship is not readily assessed, but it is fundamental since it influences people's willingness to walk in a given place [21,29,31]. From this perspective, the subjective/perceptual factors should be assessed when planning urban spaces in order to contribute to designing more sustainable cities [28,29,32].

While past research has fully addressed the technical side of measuring and representing walkability [33,34], focusing on the objective aspects (e.g., the width and height of sidewalks), there is still a wide gap in our knowledge about how urban planning copes with the subjective aspects of walkability (i.e., the comfort of walking a road).

Based on a case study research method [35], the aim of this paper is to contribute to filling this gap by proposing a multi-methodological assessment framework able to jointly assess the objective and subjective dimensions of walkability with a view to guiding future sustainable, resilient urban development.

The multi-methodological assessment framework can be a useful tool for dealing with the issue of sustainable mobility in an approach that sees walkability as a factor in the city's sustainability and growth [32].

The study consisted of several interactive steps [36]. First, we carried out a literature review to understand the different nuances involved in walkability and the most commonly used assessment methods. Second, on the basis of the results of the literature review, we identified the main indexes and indicators for measuring the aspects of walkability. Third, we used surveys [37] to empirically test the validity of these indexes and indicators, and we aggregated the results through statistical analyses. Lastly, we performed a spatial evaluation [38,39] based on Geographic Information Systems (GIS) in order to assess the geographical representation of the indicators [34].

The remainder of the paper is organized as follows. Section 2 frames the case study research method by introducing the case study; Section 3 describes the main methods applied to assess walkability; Section 4 presents the development of the assessment framework. Lastly, Section 5 discuss the model's strengths and weaknesses, as well as future directions for research.

## 2. The Case Study Research Method

In order to properly develop a multi-methodological framework able to analyze both the objective and the subjective aspects of walkability, it was decided to apply a case study research method [34,40,41], which implies the in-depth investigation of a single individual or multiple events to explore the causes of underlying general principles.

Accordingly, the case study research method involves the identification of a case study, the collection and analysis of data, and the representation of the results obtained [42]. Through this method, it is possible to open up new directions for future research. In this perspective, walkability is thus assessed in a real setting.

In a case study research, selecting an appropriate case is fundamental, since a poor choice could place the entire development of the assessment method at risk [35]. Consequently, before making our choice, we listed a number of characteristics the case study should have in order to be suitable for our purpose. First, since the walkability assessment changes according to the territorial scale of analysis, an intermediate territorial scale similar to a district would provide insight into a manageable territory and would be scalable to larger and smaller areas [39].

Second, the case needed to be a public space frequented by large numbers of people so that subjective data could be collected from the area's users. Lastly, the area had to be familiar to the researchers in order to avoid lengthening the time spent in data retrieval. In view of these requirements, the choice fell to the main university campus of the Politecnico di Torino (PoliTO, Italy), hereunder referred to as the PoliTO campus.

*The Case Study: Main Campus of the Politecnico di Torino*

From the perspective of case study research, a university campus provides fertile ground for studying and assessing various aspects of sustainability and resilience, raising awareness among students, lecturers, and administrative staff about crucial issues of our times [43]. Here, it is possible to conduct research, undertake multidisciplinary collaborations and implement sustainability solutions that can be generalized in the future. At the same time, a university campus is comparable to an urban district in terms of size and dynamics [44].

The PoliTO campus was suitable for our purpose since it hosts the university's main activities and is used by a large numbers of students, teachers, and administrative staff. Moreover, the main campus has extensive open spaces that can only be used by pedestrians, which is an important consideration. Lastly, as indicated by the PoliTO Masterplan [45], the campus is poised to begin a new season of change in terms of growth, interaction with the territory, internationalization, and sustainable planning.

The PoliTO Masterplan, managed by a selected team of designers and experts, envisages a series of projects to increase the livability of campus spaces and the provision of services. With regard to the enhancement of open spaces, the Masterplan aims to deploy coordinated actions to create new paths, green areas and places for collective activities.

Figure 1 shows the case study area, which includes the PoliTO campus (green border) as well as the surrounding area (red border).

In talking about walkability, it is essential to think about the campus's accessibility, taking intermodality into account to consider the different modes of transport available to users. Moreover, it is important to consider that, in accordance with the "last mile theory" [46], in any communication network, the last mile is more likely to reach customers and is therefore the most reasonable area to consider in a study.

Accordingly, the case study (Figure 1) includes not simply the campus but a wider area comprising as much local public transport as possible and the main railway station of Turin (the Porta Susa intermodal station).

Figures 2 and 3 show several routes on the PoliTO campus and in the surrounding area that feature differences in walkability. In fact, a first empirical observation of the study area indicated that some routes involve more challenges for the pedestrian than others.

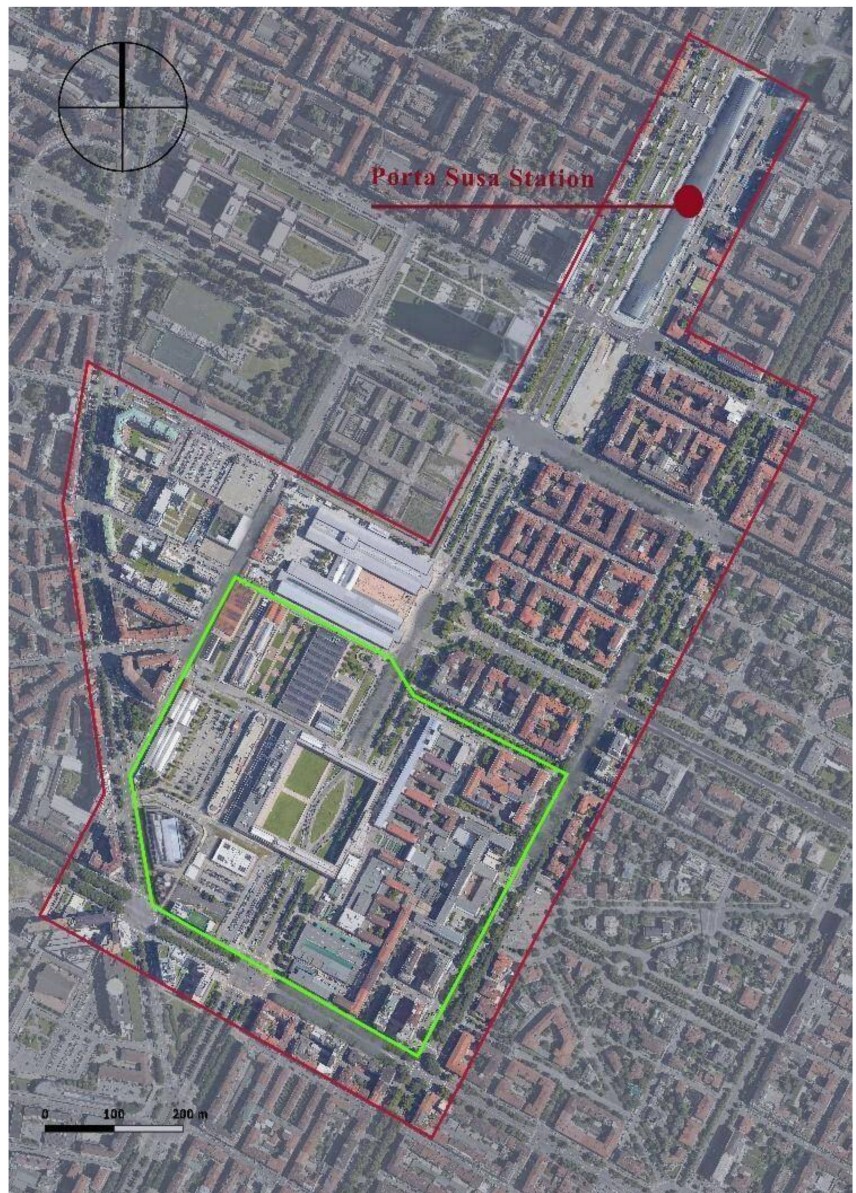

**Figure 1.** The case study area.

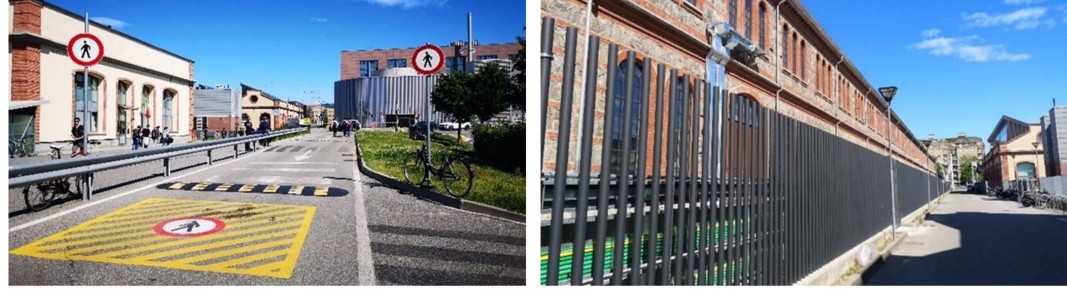

**Figure 2.** Pedestrian routes on the Politecnico di Torino (PoliTO) campus (Source: authors' photos).

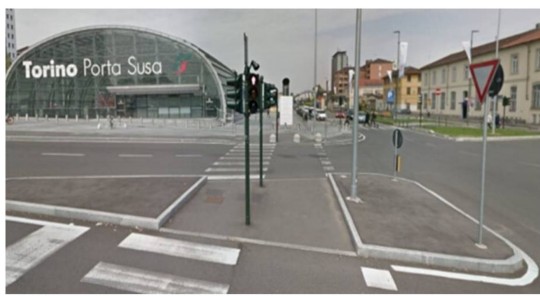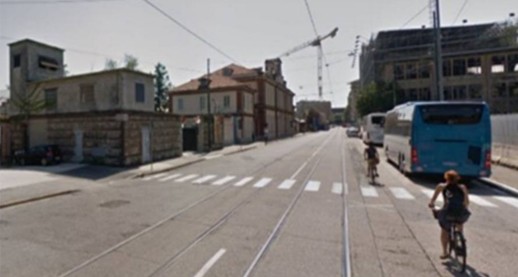

**Figure 3.** Pedestrian routes in the area surrounding the PoliTO campus (Source: authors' photo).

The photographs in Figure 2 show two pedestrian routes that cross roadways on the PoliTO campus. Traffic signs and road markings regulate the one on the left, while the other path is devoid of signage and separation.

The photo on the left in Figure 3 shows the pedestrian crossing regulated by traffic lights leading to the Porta Susa station, while the photo on the right shows a dangerous pedestrian crossing on a linear stretch without signs or traffic lights.

## 3. Research Design and Data Analysis

After selecting the case study, we specified several requirements for the multi-methodological assessment framework.

The framework should be:

(1) Able to consider objective and subjective elements of walkability;
(2) Able to quantify and measure subjective elements;
(3) Mathematically robust and sensible;
(4) Flexible and adaptable. e.g., usable at different territorial scales;
(5) Able to support the urban planning design decision-making processes.

**Table 1.** Synthesis of the multi-methodological framework.

| Phases | Steps | Activities | Results |
|---|---|---|---|
| Choice | Literature review | Definition of the keywords/search parameters<br>Definition of the time span<br>Database search<br><br>Analysis of the 16 papers selected | Selection of 16 papers containing qualitative/quantitative assessment methods to measure walkability<br>Identification of the most used indexes and indicators (4 indexes and 18 indicators) |
| Analysis | Empirical investigation | Validation of the results of the Choice phase<br>Selection of a preliminary sample to deliver the survey test<br>Validation of the reliability of the survey on the preliminary sample<br>Choice of a final sample to deliver the survey<br>Delivery of the survey to the final sample<br>Statistical analysis | Elaboration of a survey test<br><br>40 students of the PoliTO campus + PoliTO masterplan Changing the indexes and the indicators selected in the Choice phase 4 indexes and 28 indicators)<br><br>100 PoliTO users<br><br>Definition of the weights to be attributed to the indexes and the indicators |
| Evaluation | Spatial evaluation | Use of Geographic Information Systems (GIS) measures to spatialize the indexes and indicators<br>Identification of the problem areas from a walkability perspective in the study area | Spatialization of the indexes and the indicators<br><br>Suggestions for improvement in terms of walkability to support the PoliTO Masterplan |

To satisfy these requirements, the proposed multi-methodological assessment framework is organized in three phases and several steps (Table 1) in an interactive and iterative process in order to achieve solid results [47,48].

According to Table 1, the multi-methodological assessment framework is structured as follows:

(a)    Choice phase, where indexes and indicators were preliminary chosen through an in-depth analysis of the literature. First, we selected three keywords to compose the string search viz., walkability + walkability measure + walkability indicators. Second, the string has been inserted in both Scopus and Google Scholar databases to identify scientific papers in the timespan 2000–2019 (Figure 4). This research has given rise to numerous papers. Third, basing on abstract and keywords, we selected only the papers that appeared in both databases and simultaneously related to the 3 subject areas of interest: urban planning, urban planning measure and qualitative/and quantitative assessment methods). This systematic literature review provided 16 (Table 2).

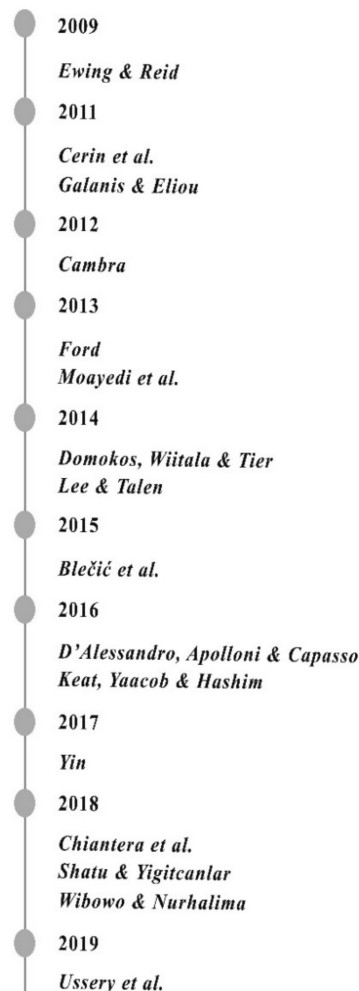

**Figure 4.** Distribution of the 16 papers in the timespan of the research.

Figure 4 shows that, although the timespan is related to 19 years, it is only since 2009 that we found scientific papers actually corresponding to our research interests. This underlines the topicality of the walkability measurement from an urban planning perspective.

**Table 2.** Quantitative and qualitative assessment methods in the 16 papers analyzed.

| Papers | Quantitative Methods | | Qualitative Methods | | |
| --- | --- | --- | --- | --- | --- |
| | Weighting of Indexes and Indicators | Statistical Analysis | Empirical Investigation | Assessment Survey | Visualization through GIS and CAD (Computer-Aided Drafting) Tools |
| Ewing and Handy, 2009 | | ✔ | ✔ | | |
| Cerin et al., 2011 | | ✔ | | ✔ | |
| Galanis and Eliou, 2011 | | ✔ | | | ✔ |
| Cambra, 2012 | ✔ | ✔ | | | |
| Ford, 2013 | ✔ | | | | |
| Moayedi et al., 2013 | ✔ | | | | |
| Domokos, Wiitala and Tier, 2014 | | | ✔ | ✔ | |
| Lee and Talen, 2014 | | ✔ | ✔ | | ✔ |
| Blečić et al., 2014 | ✔ | | ✔ | | |
| D'Alessandro, Apolloni and Capasso, 2016 | ✔ | | | | |
| Keat, Yaacob and Hashim, 2016 | | | ✔ | ✔ | |
| Yin, 2017 | | ✔ | ✔ | | |
| Chiantera et al., 2018 | ✔ | | | | ✔ |
| Shatu and Yigitcanlar, 2018 | | ✔ | | ✔ | |
| Wibowo and Nurhalima, 2018 | | | ✔ | | |
| Ussery et al., 2019 | | ✔ | | | |

In Table 2, the 16 papers are summarized according to the assessment method used. According to the literature, the main quantitative methods are:

(1) Weighting indexes and indicators to produce a global index. Indexes and indicators are chosen by researchers on the basis of the literature or empirical analyses. The method is very flexible and can be applied at several territorial scales [39];

(2) Statistical analyses, which provide a robust evaluation by using highly objective analytical attributes such as averages, maximum and minimum values, correlation, and agreement coefficients and standard deviation [49];

The main qualitative methods are:

(1) Empirical investigation, which can assess both measurable and perceptual elements by direct observation in the analyzed area [50].

(2) Assessment survey, which aims to capture the subjective aspects of a problem. The difficulty here lies in selecting the correct survey structure [37];

(3) Visualization through GIS [50] and CAD tools [34] to visualize the current state of the study area and to represent future scenarios.

The literature review indicated that the choice of one method rather than another depends on two main factors: the geographical scale of the analysis and the purpose of the assessment. Currently, it is very difficult to identify an assessment method that is suitable for every situation in a multi-scale perspective [27].

Nevertheless, researchers have begun to advance proposals for overcoming these problems. The most widely used solution is to combine qualitative/quantitative assessment methods [47,51] in order to include both aspects of the assessment while making the method more flexible. However, there are still few studies that propose all the assessment methods simultaneously (Table 2). For example, statistical analyses are often employed after all the other assessment methods to verify the robustness of the results [28,34], while assessment surveys are usually used before weighting indexes and indicators to gauge the level of satisfaction with the indicators [52].

(b) Analysis phase, consisting of an empirical investigation of the case study area and a survey administered to the main categories of PoliTO campus users. In order to verify the reliability of the survey, a preliminary test was made on a sample of 40 students. Subsequently, survey data

were analyzed using different statistical techniques. Through the survey, the results of the Choice phase were tested, making changes and enriching it with data, thus making the model more robust and objective;

(c)  Evaluation phase, where the current status of the PoliTO campus was assessed. This phase employed a GIS software application called Quantum-Geographic Information System (QGIS) [53] to assess potential associations between a number of built environment characteristics and walking [54] and to have a visual representation of the evaluation problem [55]. Among the many available visualization tools [39,50,56], we decided to use QGIS [53] since it is an open-source software system that does not require a license, uses readily consulted open data, and georeferences objects to be assessed on any geographic scale (city, neighborhood, or single street), providing easy-to-read output. Moreover, it is widely used, making the method presented here easily replicable.

In general, using visualization tools can promote a shared understanding among the stakeholders involved in a decision process [57–60] and is useful in complex problems such as walkability, which involve many different stakeholders and aspects.

Thus, the QGIS tool in the third phase (Table 1) contributes to the assessment by helping stakeholders to "get on the same page" [61] and to have a collective insight [62] about the issues involved.

## 4. Findings

### 4.1. Choice Phase

To identify the main walkability indexes and indicators to be used in the proposed multi-methodological assessment framework, we studied and analyzed the 16 papers selected through the literature review discussed in Section 3. The analysis yielded 18 indicators divided into 4 indexes (Table 3).

**Table 3.** Indexes and indicators resulting from the literature review.

| Indexes | Indicators | References |
|---|---|---|
| Security | Presence of intersections<br>Drivable speed<br>Existence of conflict area between pedestrian and vehicular traffic<br>Types of roads | [33,39,52,63–66] |
| Quality of routes | Sidewalk's length<br>Condition of the pavement<br>Non-sliding paths (with obstacles)<br>Well connected<br>Slope | [21,28,29,31,33,34,39,52,63–66] |
| Intermodality | Presence and coverage of public transport stops<br>Cycling | [31,63] |
| Comfort | Presence of trees/meadows<br>Adequate lighting<br>Possibility of stopping due to benches<br>Architectural variety<br>Buildings with monotonous colors<br>Possibility to see the continuity of the route<br>Presence of commercial activity | [21,29,31,33,37,39,49,50,52,63–66] |

As shown in Table 3, according to the revised literature, the most commonly used indexes for assessing walkability are Security, Quality of route, Comfort, and Intermodality. Moreover, each index can in turn be measured with different indicators. In detail, the Security index can be measured through 4 indicators (7 papers), the Quality of routes contains 5 indicators, the Intermodality index contains 2 indicators (2 papers), and Comfort has 7 indicators (12 papers). Unsurprisingly, the majority of the indicators refer to the Quality of routes and Comfort indexes. This is probably because it is difficult to identify general indicators capable of measuring these indexes' high subjectivity.

*4.2. Analysis Phase*

The first step of the analysis phase was an empirical investigation of the PoliTO campus (Table 3) to determine which of the indexes and indicators found through the literature review were most appropriate (Table 1). For this purpose, we interviewed a first sample consisting of the PoliTO Masterplan Team together with a selected group of 40 students. They were asked to analyze the indexes and indicators shown in Table 3 in terms of their applicability to the PoliTO campus. The interviewees found that the Security, Quality of routes, Comfort, and Intermodality indexes perfectly fit the PoliTO campus's needs. By contrast, the indicators found in the literature review were too generic to correctly assess the current situation of the PoliTO campus or for use in planning projects. Accordingly, each indicator in Table 3 was further specified to better reflect the case study's needs. For instance, the "cycling" indicator in the Intermodality index (Table 3), which referred simply to the presence of cycle paths, was divided into two indicators, namely "parking spaces for own bike" and "bike sharing stations" (Table 4) denoting that the area in question has provision for users to park their own bikes and features bike sharing stations (Figure 1).

**Table 4.** Indexes and indicators selected in the analysis phase.

| Indexes | | Indicators |
|---|---|---|
| **Security** | | Presence of busy roads |
| | | Traffic light pedestrian crossings with sufficient time |
| | | Non-lighted pedestrian crossings in neighborhood streets |
| | | Separation of pedestrian/cycling/cable/accessible routes |
| **Quality of routes** | Internal | Tightening of sidewalk |
| | | Condition of the pavement |
| | | Non-sliding paths (with obstacles) |
| | | Well connected with the outside |
| | | Slope |
| | External | Tightening of sidewalk |
| | | Condition of the pavement |
| | | Non-sliding paths (with obstacles) |
| **Intermodality** | | Parking spaces for own bike |
| | | Easy accessibility by public transport |
| | | Own car parks |
| | | Bike sharing stations |
| | | Car sharing stations |
| **Comfort** | | Acoustic pollution |
| | | Covered routes |
| | | Presence of trees/meadows |
| | | Presence of baskets |
| | | Adequate lighting during night/evening hours |
| | | Possibility of stopping due to the presence of benches |
| | | Presence of water points |
| | | Presence of tall buildings |
| | | Buildings with monotonous colors |
| | | Possibility to see the continuity of the route |
| | | Refreshment points of the PoliTO campus |
| | | Study points in the PoliTO campus |
| | | Spaces where crowding is created in PoliTO campus |
| | | Spaces where crowding is created outside PoliTO campus |

In addition, the indicators for the Quality of route index were split into two macro categories, internal and external, to better analyze the situation on and off campus (see green and red borders in Figure 1). With the same rationale, some indicators were eliminated from the analysis:

the "presence of commercial activity" indicator in the Comfort index was considered unnecessary for the PoliTO campus.

The 28 indicators resulting from the empirical investigation are listed in Table 4.

As shown in Table 4, the final selected indexes are: Security, Quality of routes, Intermodality, and Comfort. Despite the changes made, according to the first interviewed sample, we have classified the indicators into indexes based on the analysis of the literature as for example: the indicator "tightening of the sidewalk" has been associated with the Quality of routes index because it refers to the specific structural characteristics of the pedestrian area.

The second step of the analysis phase used surveys (see in Supplementary Materials) and statistical analyses to test the sensitivity of the indexes and indicators (Table 4) and understand the weights of each index and indicator in the users' subjective perceptions.

The surveys consisted of 36 closed questions in order to facilitate completion and reduce the dispersion of responses [67]. Respondents were asked to rate their agreement with each question on a 5-point Likert scale [68,69] ranging from 1 (strongly disagree) to 5 (strongly agree).

A sample item is reported in Table 5.

**Table 5.** Example of closed question provided in the questionnaire

| *There are many busy roads in the area inside the PoliTO campus with heavy vehicular traffic.* | | | | | | |
|---|---|---|---|---|---|---|
| **Strongly Disagree** | 1 | 2 | 3 | 4 | 5 | **Strongly Agree** |

After an internal test to verify the reliability of the survey structure, the surveys have been sent by e-mail to the daily employees and users of the PoliTO campus including students (from 4 master's degrees), faculty members (professors, researchers, and research fellows) and technical/administrative staff. The completed surveys collected have been 100. This sample size corresponds to the non-statistical sampling method called "judgmental sampling" [70], according to which the choice is entrusted to the researcher with criteria of representativeness and convenience: the increase of the empirical basis ends when the addition could give a null contribution [70].

Based on the 100 surveys, statistical analysis was performed in order to analyze the data and assign weights for indexes and indicators as suggested by the literature [33]. After applying several simple statistical analyses including mode, arithmetic mean, weighted average, and standard deviation [71], we decided to focus on calculating the weighted average since it is better able to reflect the priorities of the users' real preferences by assigning each value its own degree of importance, producing more sensitive results [52]. In fact, according to the definition of the weighted average, the values in analysis are summed, each multiplied by a coefficient that defines their "importance" and the result is divided by the sum of the weights [71]. The aggregated weights of the indexes and of the indicators obtained through the weighted average are shown in Table 4.

As shown in Table 6, Security is considered the most important index (29%) and also contains the most important indicator, e.g., "presence of busy roads" (31%), given that from the individual's point of view, being able to walk in a safe place is an extremely important aspect of sustainable public space planning. Almost equal to the first index is the Quality of routes (28%), whose most important indicator is "non-sliding paths" (15%). This highlights the importance of eliminating architectural barriers in public spaces. The third index is Intermodality (22%), where the "easy accessibility by public transport" indicator is emphasized (23%). Public transport is widely used in Turin. Most users of the PoliTO campus reach it by train or bus rather than bikes or private transport. This means that being able to reach a train or bus station quickly is very important. The Comfort index came last in the rankings (22%). This does not mean that comfort is not an important aspect for the PoliTO campus, but that any associated problems have to a certain extent been solved. Currently, users perceive the PoliTO campus as comfortable, except for a problem highlighted by the "spaces where

crowding is created in PoliTO campus" indicator (10%) which reflects the need for a more rational and planned use of space to avoid overcrowding.

**Table 6.** Indexes and indicators with weights obtained in the analysis phase.

| Indexes | Weights of Indexes | Indicators | | Weights of Indicators | |
|---|---|---|---|---|---|
| Security | **29%** | | Presence of busy roads | 31% | Minimize |
| | | | Traffic light pedestrian crossings with sufficient time | 23% | Maximize |
| | | | Non-lighted pedestrian crossings in neighborhood streets | 19% | Minimize |
| | | | Separation of pedestrian/cycling/cable/accessible routes | 26% | Maximize |
| Quality of routes | 28% | Internal | Tightening of sidewalk | 12% | Minimize |
| | | | Condition of the pavement | 13% | Maximize |
| | | | Non-sliding paths (with obstacles) | 15% | Minimize |
| | | | Well connected with the outside | 12% | Maximize |
| | | | Slope | 11% | Minimize |
| | | External | Tightening of sidewalk | 13% | Minimize |
| | | | Condition of the pavement | 12% | Maximize |
| | | | Non-sliding paths (with obstacles) | 12% | Maximize |
| Intermodality | 22% | | Parking spaces for own bike | 20% | Maximize |
| | | | Easy accessibility by public transport | 23% | Maximize |
| | | | Own car parks | 17% | Maximize |
| | | | Bike sharing stations | 21% | Maximize |
| | | | Car sharing stations | 19% | Maximize |
| Comfort | 21% | | Acoustic pollution | 8% | Minimize |
| | | | Covered routes | 5% | Maximize |
| | | | Presence of trees/meadows | 6% | Maximize |
| | | | Presence of baskets | 7% | Maximize |
| | | | Adequate lighting during night/evening hours | 7% | Maximize |
| | | | Possibility of stopping due to the presence of benches | 6% | Maximize |
| | | | Presence of water points | 6% | Maximize |
| | | | Presence of tall buildings | 8% | Maximize |
| | | | Buildings with monotonous colors | 8% | Minimize |
| | | | Possibility to see the continuity of the route | 7% | Maximize |
| | | | Refreshment points of the PoliTO campus | 8% | Maximize |
| | | | Study points in the PoliTO campus | 7% | Maximize |
| | | | Spaces where crowding is created in PoliTO campus | 10% | Minimize |
| | | | Spaces where crowding is created outside PoliTO campus | 8% | Minimize |

Moreover, while some indicators refer to positive elements from the walkability point of view, others are negative. Therefore, the corresponding weights should be minimized or maximized (Table 4) depending on these characteristics (e.g., "presence of busy roads" refers to a negative characteristic therefore should be minimized, as opposed to "traffic light pedestrian crossings with sufficient time" that should be maximized—Table 6).

### 4.3. Evaluation Phase: QGIS Measure

The Evaluation phase involves the visual representation of the objective/technical and subjective/perceptual elements of walkability. This phase employed QGIS and sought to provide a complete picture of the current walkability situation on the PoliTO campus.

To this end, the 28 indicators found in the literature and weighted through the surveys (Table 4) were first georeferenced. Here, some indicators required special attention. For example, the "bike sharing stations" indicator cannot be represented in a single way, since the footprint of each bike sharing station differs according to the number of stalls. Similarly, we decided to georeference "adequate lighting during night/evening hours" by representing the footprint of the light cast by the lamps.

Lastly, georeferencing the "non-lighted pedestrian crossings in neighborhood streets" indicator proved particularly problematic. In fact, according to the Torino Sustainable Urban Mobility Plan (PUMS) [72] and a site inspection (Figure 1), some non-lighted pedestrian crossings in the area in question cannot be considered dangerous, as alternative routes such as pedestrian overpasses are provided for crossing the roads. A different level of danger was thus assigned to each non-lighted pedestrian crossing on the basis of the information provided by the PUMS.

Georeferencing all the indicators with QGIS created a number of shape files, which are vector drawings using geometric shapes [73]. The shape files were then converted into raster maps [73], which are digital drawings that can store different kinds of data.

Punctual, linear, and spatial raster were produced, depending on the footprint and the spatial distribution of each indicator (Figure 5). This step resulted in 28 raster maps, one for each indicator.

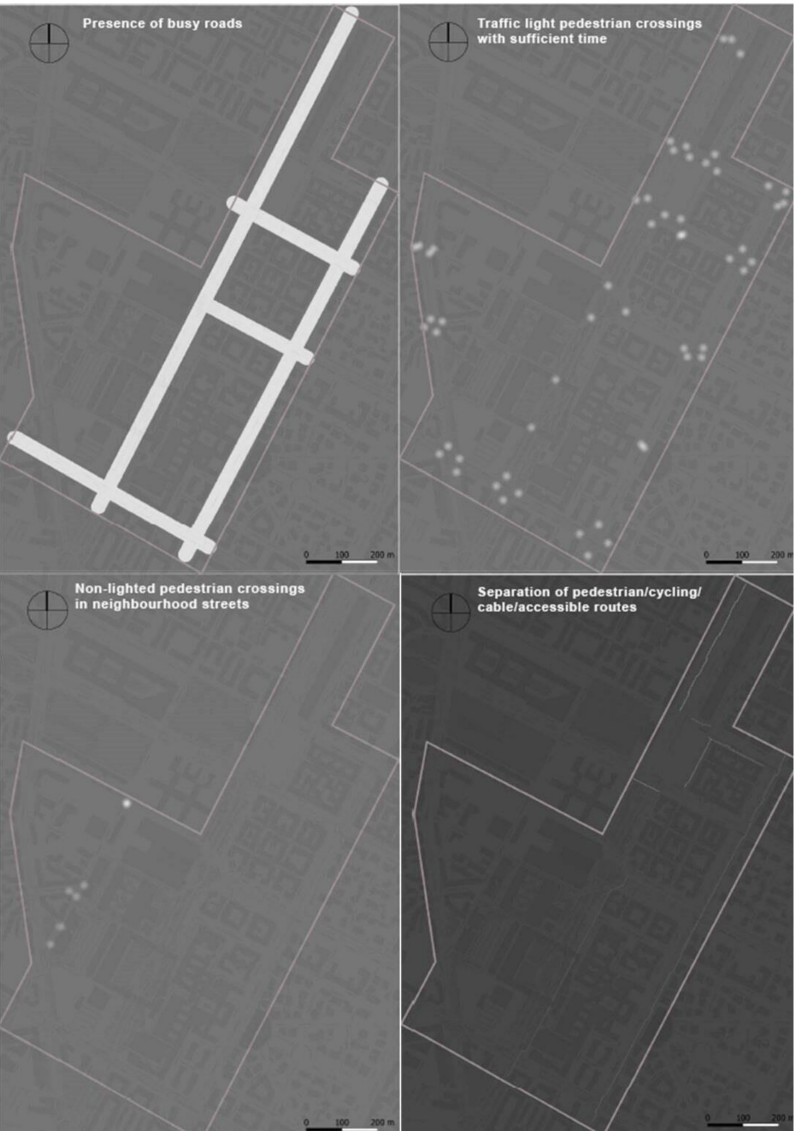

**Figure 5.** Examples of raster maps (Security index).

Figure 5 shows examples of raster maps for the Security index indicators: "presence of busy roads" and "separation of pedestrian/cycling/cable/accessible routes" maps use a linear raster created using the QGIS "Rasterize from vector to raster" tool. The "traffic light pedestrian crossings with sufficient time" and the "non-lighted pedestrian crossings in neighborhood streets" maps use punctual raster, which have been spatialized using the QGIS Kernel Density Estimation (KDE) tool [39]. This tool reports the diffusion of a phenomenon in a circular point with a radius defined appropriately according to the phenomenon represented [39] (e.g., for the indicator "traffic light pedestrian crossings with sufficient time" a radius of 20 m was used, considering it appropriate according to the phenomenon analyzed).

Moreover, the "adequate lighting during night/evening hours" indicator was an exception and it is represented by a spatial raster, due to the fact that it was important to highlight the streetlamp's different levels of light ray refraction (RN) using a spatial buffer to better highlight the differentiation of some areas compared to others (Figure 6).

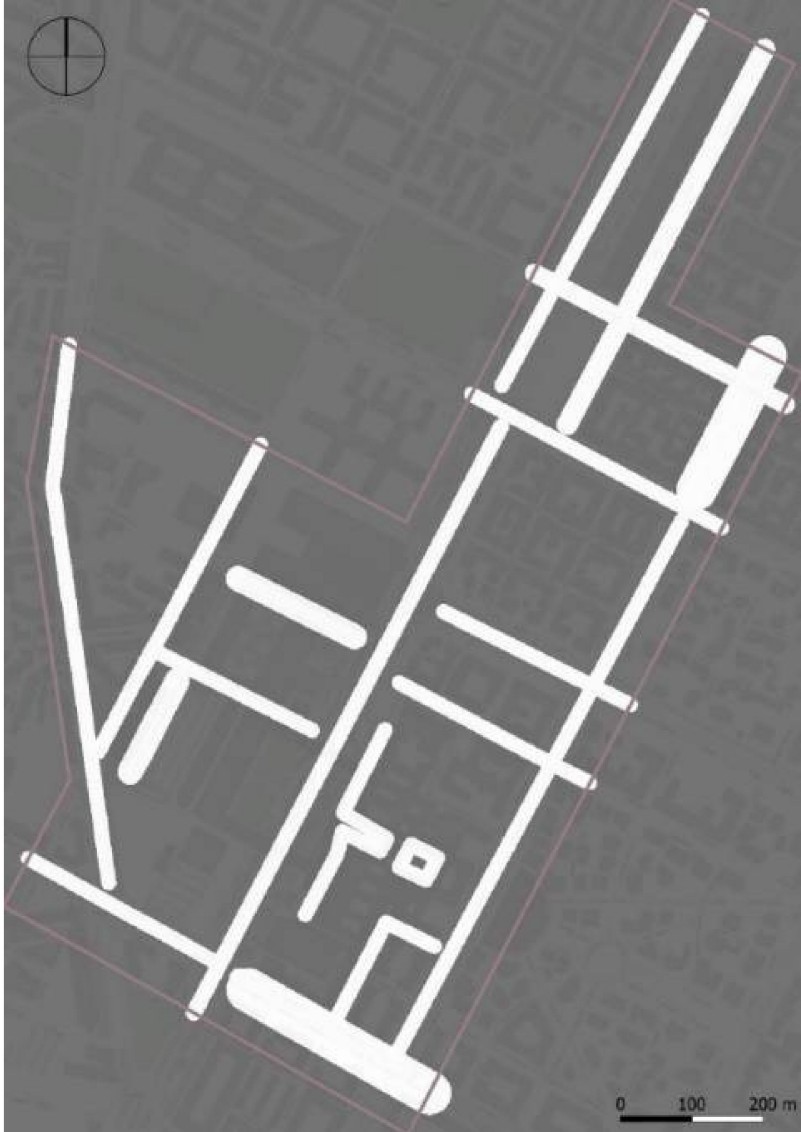

**Figure 6.** Raster map of the "adequate lighting during night/evening hours" indicator (Comfort index).

As Figure 6 shows, for this indicator we first made a buffer with the QGIS "Variable Distance Buffer" tool using a width based on the different levels of light ray reflection. Then, we generated the raster with the QGIS "Rasterize" tool.

Each raster map was assigned the weight established for the associated indicator in the previous phase (Table 5). All indicators referring to a specific index were then summed to produce four cost raster maps, which represent the cost in terms of walkability of traveling through a certain route [39].

Figure 7 shows the cost raster map for the Security index. The red areas are those in which it is less safe or pleasant to walk.

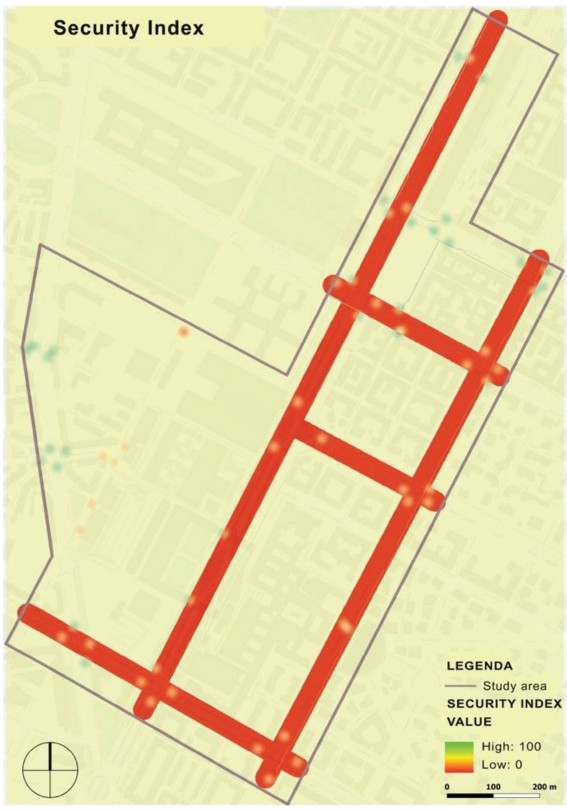

**Figure 7.** Example of a cost raster map (Security index).

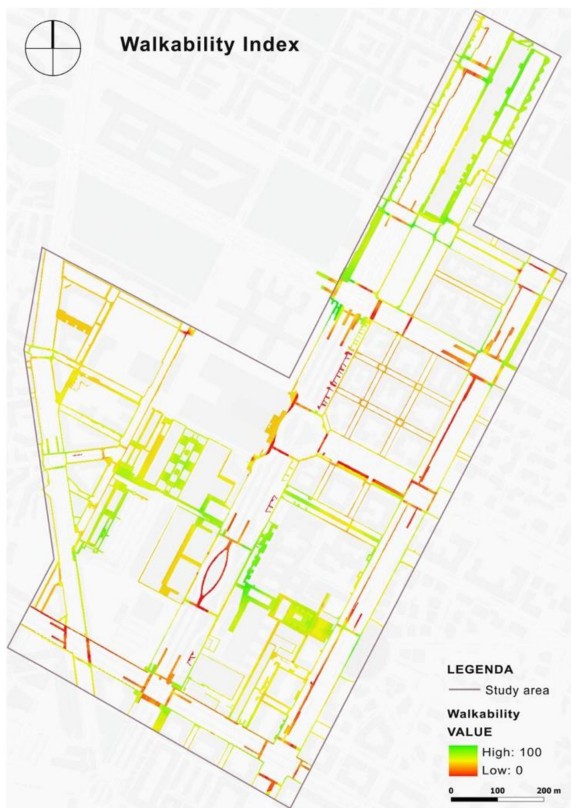

**Figure 8.** Walkability index.

After producing a cost raster map for each index, the overall weights of the indexes were considered (Table 5) and inserted in the analysis. Each cost raster map was assigned the weight of each index, and the maps were then summed to produce the final cost raster map, also called walkability index [39]. Unlike the previous maps, the walkability index represents only the pedestrian areas (Figure 8).

In Figure 8, the red areas are the most problematic in terms of walkability, considering all four indexes together, while the green areas are the most walkable one.

## 5. Discussion

Through the application of this multi-methodological assessment framework, we are better situated to provide some initial reflections about the indexes and indicators used as well as about the raster maps, highlighting how some elements could affect the improvement of the quality of walkability while also having a positive impact in relation to urban resilience. This is the case, for example, of maximizing the "covered routes" indicator (Table 5), since implementing shading is essential both to create more comfortable spaces for walking and to contribute to the reduction of heat islands in terms of resilience [11,74].

In details, Figure 8 allows to draw an overall picture of the critical issues related to walkability, which can be mostly analyzed in detail through the cost raster maps of each index (Figure 7).

Accordingly, the cost raster map of the Security index (Figure 7) clearly shows some critical values of some indicators applied to the study area (Figure 1). Figure 7 highlights the indicator "Presence of busy roads" that disturb the usability of users who reach the PoliTO campus (red lines), together with the "Non-lighted pedestrian crossing in neighborhood streets" (red point). The red dot therefore highlights a pedestrian crossing without traffic lights in the area under investigation. Although the intersection is not located on a road classified as busy, it still constitutes a danger, because it is also an important junction point for pedestrian flows that reach the PoliTO campus from the north of Turin.

In agreement, the aforementioned critical pedestrian crossing appears particularly evident also in the Walkability Index map (Figure 8), bringing the attention to an area that is generally considered quite good in terms of walkability (yellow areas).

The analysis of the cost raster maps of each index jointly with the overall Walkability index map allows to study future design solutions in order to mitigate the current negative impacts, enhancing pedestrian security and the usability of walking space.

The study and the observation of the raster maps has brought to light critical morphological aspects that can be corrected in order to design a more resilient environment, which can contemplate solutions attentive to individual specificities by enhancing the use of the roads as a public space [75].

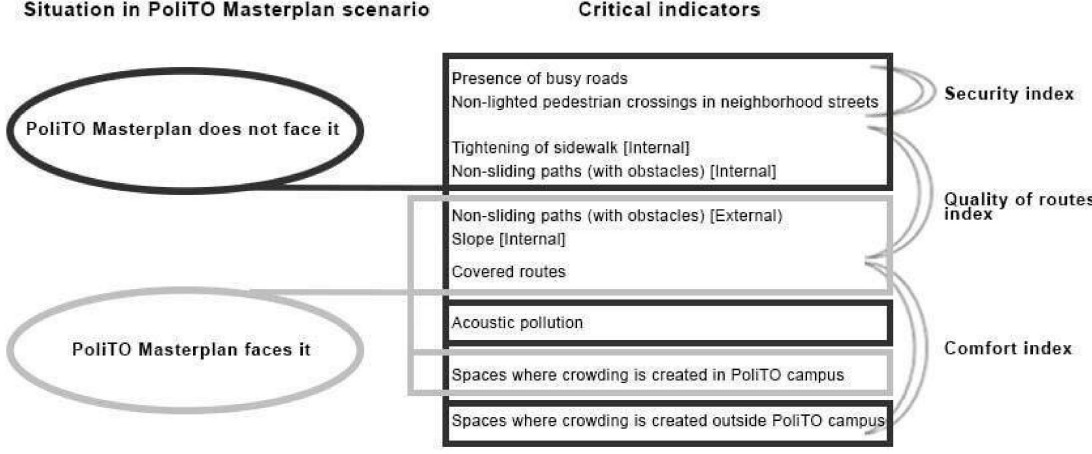

**Figure 9.** Most critical indicators and current situation in the PoliTO Masterplan.

Moreover, the value of the maps does not only concern morphological aspects, but also allows to highlight sociological elements. This is the case of the indicator "Spaces where crowding is created outside PoliTO campus" resulted critical (Figure 9) in the analysis, underlying the users' discomfort in walking to the PoliTO campus. With the same reflection, the indicator "Adequate lighting during night/evening hours" is not a critical element (Table 5), pointing out a minimum social discomfort in walking during night/evening hours.

Starting from those reflections, we analyzed in depth the ongoing PoliTO Masterplan process. We look at the PoliTO Masterplan documents in light of the calculated walkability index (Figure 8) and discussed them with the team in order to fully understand whether or not the most critical indicators shown in Table 4 were directly or indirectly taken into account in the Masterplan project proposals. (Figure 9).

As can be seen from Figure 9, the PoliTO Masterplan projects address 4 out of 10 critical indicators, namely: "non-sliding paths," "slopes," "covered routes," and "spaces where crowding is created in PoliTO campus." This is a strong improvement in terms of walkability and resilience of the PoliTO campus, although it is not enough for the campus to be considered totally walkable. However, it is important to underline that the PoliTO Masterplan projects are still ongoing and the PoliTO team could use/apply the results of our analysis to further improve the PoliTO campus situation. Moreover, some of the aforementioned critical indicators are not currently a responsibility of the PoliTO Masterplan being concentrated in areas outside the campus and therefore managed by different subjects. This is the case of the indicators "presence of busy road," "non-lighted pedestrian crossing," and "spaces where crowded is created outside PoliTO campus."

## 6. Conclusions and Future Developments

This paper analyzed one case study dealing with resilient urban planning aiming to understand the possible contribution of walkability assessment. In this section, we summarize our answers to the research question we formulated in the introduction: Is it possible to design a multi-methodological assessment framework able to jointly assess the objective and subjective dimensions of walkability?

The case study deals with a university campus in Italy (PoliTO), allowing to investigate various aspects of sustainability, resilience, and walkability. Concerning our research questions, we could report that:

1. The Masterplan addresses the issue of walkability indirectly, namely it is not explicitly mentioned in the documents;
2. Among the 10 critical indicators identified by our framework, the Masterplan projects address 4 of them ("non-sliding paths," "slopes," "covered routes," and "spaces where crowding is created in PoliTO campus"), showing particular attention to the morphology of the pedestrian streets, an attitude quite consistent with the training of the experts who drafted the Masterplan;
3. The PoliTO Masterplan Team is determining whether the Masterplan's scope can be broadened to reflect the findings that emerged from applying the multi-methodological assessment framework presented here. The idea is to be able to include roads and sidewalks around the PoliTO campus since they have a significant impact on its accessibility and walkability.

Thanks to the analysis of the above case study and the strong literature review, we have tested that the multi-methodological assessment framework is functional in terms of scientific robustness and flexibility, given its combined use of hard (quantitative) and soft (qualitative) assessment methodologies [76]. This combination provided the study with the solid underpinnings needed to take an integrated approach to elements belonging to different decisional domains and to apply the model at different scales. It is worth underlining that, as it is organized in successive interactive/iterative phases, the proposed framework is flexible: each phase can be seen as the basis for subsequent or previous phases, so that the process can be re-thought as new or more accurate information becomes available.



In terms of completeness, the multi-methodological assessment framework contributes to overcoming the idea that objective and subjective aspects are "not part of the same planning project" [77]. Thanks to the combination of hard and soft methods, the framework can consider objective (physical) and subjective (perceptual) aspects simultaneously and represent them visually using GIS. It can thus provide easily readable results that can be applied in establishing guidelines [78] for future plans and projects.

With regard to the type of contribution that walkability assessment can provide to resilient urban planning, it has been pointed out that public space planning and walkability are intertwined in a relationship of non-negligible causality: each one involves and enhances the other, adding psychological well-being, aesthetic pleasure, promoting social exchanges or simply spending free time outdoors. Correct walkability planning is an essential part of planning sustainable cities, as it controls the way people move and determines the way they will move in the future [21,79]. In this perspective, walkability assessment can be part of a planning process, useful in understanding all its phases: from the current status to the planning proposals, up to the design of possible future scenarios [80].

It should be emphasized that the multi-methodological assessment framework presented here leaves room for future developments. In future work, we plan to verify how the indicators would change and what dynamics would be involved when a wider territorial scale is considered. Moreover, it would be interesting to carry out surveys on the "intermodality" index in greater depth by including analyses about users' movements and preferences stemming from the cost of the trip, not only in terms of money and time. Results could thus be organized in relation to users' preferences, according to more specific indicators that better frame the situation of the Intermodality index.

Lastly, the proposed multi-methodological assessment framework will be tested to determine whether it can be applied not only to assess an area's current walkability status, but also to compare different project scenarios.

**Supplementary Materials:** The following are available online at http://www.mdpi.com/2071-1050/12/19/8131/s1, The Attached 1: Walkability Evaluation of the Main PoliTO Campus. Survey—Walkability evaluation in PoliTO campus consisted of 36 closed questions and was delivered to a sample of 100 users. Respondents were asked to rate their agreement with each question on a 5-point Likert scale [68,69] ranging from 1 (strongly disagree) to 5 (strongly agree). This survey was used during the Analysis phase (Section 4.2).

**Author Contributions:** Conceptualization and methodology F.A. and I.M.L; software—methodology L.L.R.; software—maps, L.L.R. and M.G.; writing—original draft, M.G.; writing—review and editing, F.A. and I.M.L. All authors have read and agreed to the published version of the manuscript.

**Funding:** This research received no external funding.

**Acknowledgments:** The authors would like to thank the Politecnico di Torino Masterplan Project Team and, in particular, Caterina Barioglio. We also thank Simona Fiorino for her contribution to this study.

**Conflicts of Interest:** The authors declare no conflict of interest.

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
