# Peer review of "Supporting Resilient Urban Planning through Walkability Assessment"

_sustainability, doi:10.3390/su12198131_

Round 1
Reviewer 1 Report
The study examines the urban resilience in terms of walkability aspect in Italy. It seems an interesting paper though I have serious reservations for fundamental aspects of the study.
Comments
There are many repetitive sections in the text. For instance, the introduction (at the end of pg. 1) and literature review (at the beginning of pg. 3) (i.e., soft mobility)
The section Covid-19 in the introduction seems irrelevant (pg. 2)
Are not there any walkability studies that used mixed-methods or qualitative aspects?
The gap in the literature is not clear. Does it mean that there are not “enough” studies on walkability, particularly on the subjective aspects?
Why is a case study approach conducted by the author(s)? What makes a case study for being able to use in this study? I recommend the author(s) to check the case study method and include some core information about it.
Page 3, security, security?
Page 3. Ln 132-133, this is a very strong statement and needs to be refined
Page 4. It would be great to see a summary of the reviewed 25 literature pieces.
Also, were 25 literature studies selected based on the title, abstract, or the entire text for those 3 keywords - “walkability”, “walkability measure” and “walkability indicators”-? And why?
Page 4, Ln 177. Typo
For the case study location, I strongly believe that a study location in/around/adjacent to a campus affects the survey results in terms of socio-demography and the participants’ perception. Did the author(s) consider any control group/study location? It is emphasized because this pattern even affects the travel (in your case walkability) behavior and thus the quality of the study results.
Figure 4-9 should be of high quality as it is hard to follow the core message in these figures.
There are a lot of “(Error! Reference source not found.)” and these are important references that need to be seen.
Where is the survey question/s the one include/s 36 questions? I recommend the author(s) to create a summary of the survey questions.
How were the participants of the survey selected?
Why was the survey stopped as soon as the number of participants reached to 100?
How was the weighted average method performed? Any formula or calculation?
What are the “weights” on the left and “weight” on the right side of the Table 2? It is vague with the current representation
In the same table, it is perhaps noise pollution, not the acoustic.
How did the author(s) classify the indicators in indexes? For instance, the tightening of the sidewalk (under the quality of routes) also can be considered under the security index.
Discussion sections need to be expanded remarkably as it is the core aspect of this study.
In its current form, the paper seems like a campus-related/planning project report rather than a scientific research paper.
Overall, this is an interesting study; however, I do not see a clear association between urban resilience and walkability in this study.
Reviewer 2 Report
This paper explores walkability in cities in order to support urban resilience. The topic is interesting and timely. Here are my comments to improve the paper:
1- make the knowledge gaps and research objectives clearer in the introduction section.
2- your literature review section is rather short. Please make sure you provide more details on the relevant studies, whilst clearly identifying the knowledge gaps.
3- include these references to acknowledge the issues surrounding urban morphology:
https://doi.org/10.3390/urbansci4020023
https://doi.org/10.1007/978-3-030-12381-9_12
4- there are so many errors in your referencing in the result section.
5- you need to expand the discussion section. Perhaps you need double the content you have currently included. You need to discuss what your findings mean for policy makers. In other words, what are the implications of your work for policy.
6- you need to refer back to the existing literature and make arguments in the discussion section.
7- similarly, you need to further elaborate your conclusion section. make the key findings clearer and summarise all the main points.
8- please proofread the paper as there are several writing issues.
Reviewer 3 Report
line 131: I would change 'objective/physical and subjective/perceptual' to 'objective(physical) and subjective(perceptual)' to avoid confusion;
line 166-183: The 5 paragraphs can be combined into one paragraph.
line 281-412: please check the links of reference/table after line 281.
line317: What are the 36 questions?
Table 2: How did you obtain the weights from the surveys? What do you mean by Minimize and Maximize in the table?
Figure 5: please explain your classification of raster type. In GIS, rasters are spatial data. What are your definitions of punctual and linear raster? And,I don't think it is necessary to classify raster data here.
line 376: What is your radius in the caculation of Kernel Density in QGIS?
Although you discussed the contribution of walkability to urban resilience in the introduction, which is a good pespective, the major part of the paper--literature review, methods, results, discussion--does not really related to urban resilience. You might need to either add more things in these parts or rewrite your introduction.
Round 2
Reviewer 1 Report
The authors have made some great improvements in their resubmission. The following issues should be given additional consideration.
Reviewing 16 articles between 2000 and 2020 sounds a very few study/filtering scope. After checking the same method the author(s) used, I personally found several studies. It is noteworthy that the authors should explain their “filtering” category (i.e., 3 main categories including urban planning, measuring, qualitative and quantitative assessment methods).
Regarding the case study location, the author(s) explained well for this concern. However, after checking the survey questions, there are only 3 participant categories (Student, Teacher/Research fellow, etc., and administrative staff) and it shows a bias that I emphasized in the previous revision. Having only these 3 categories creates limited and possibly biased results as a consequence of certain socio-demographic features.
My other question from the previous revision also remains unanswered comprehensively. Were the participants selected randomly or did the author(s) follow another approach?
Stopping the survey studies as soon as reaching 100 also does not explain the concern. Also, the reference is in the Italian language and it is better to provide some other sources in English. I suggest the author(s) investigate more on the sample size selection criterion.
The discussion is still the same as the previous submission though the author(s) improved the conclusion. The discussion section needs to be expanded.
Reviewer 2 Report
Thank you for addressing the comments.
Author Response
Please see the attachmen.

Round 3
Reviewer 1 Report
Thank you for addressing my comments.
* Reference #70 (The English source for sample size selection) was not cited in the manuscript.
